# ASPIRE to a Better Future: The Impact of the Pandemic on Young People, and Options for Schools Post-COVID-19

**Sue Roffey**

Clinical, Educational and Health Psychology, University College London, London WC1H 0AP, UK; s.roffey@ucl.ac.uk

**Abstract:** Young people have, in effect, had two years of normality taken from their lives by the pandemic—and for many this has occurred at a crucial time of development. Using the ASPIRE framework of Agency, Safety, Positivity, Inclusion, Respect, and Equity, this paper explores what has happened to adolescents in the UK and elsewhere, the impact this may have had on identity formation and establishing a meaningful sense of self, feelings of belonging and safety, mental health, hope for the future, and relationships. Students already facing disadvantages and adversity have been particularly hard hit. School attendance levels are falling, giving rise to further concerns about wellbeing. Referring to each principle, we discuss the options for responding in ways that support a more positive future.

**Keywords:** adolescents; identity; pandemic; mental health; the ASPIRE principles; education; relationships; wellbeing





## 1. The Life Tasks of Adolescents

Young people between the ages of 11 and 18 are at a critical age—what happens to them, the opportunities available to them, and the sense they make of the world during these years all have a lasting impact on the rest of their lives. It is the time to explore who they are, who they want to become, and what values they hold. Many do this with others of a similar age, try out new ways of being, take risks, and defy authority. This sometimes brings them into conflict with their family of origin. Parents of teenagers want to protect them and keep them safe, but young people often want increasing independence. Renegotiating family relationships is often ongoing. Relationships with peers take on a new meaning, as these become the reference point for the development of values, behaviours, and beliefs. Adolescents are also future focused, exploring options for their lives ahead including work, relationships, and aspirational endeavours.

Physical challenges include sexuality, sexual identity, body image, and keeping physically healthy. The Good Childhood Report 2022 [1] asked 44,000 children and young people in the UK about their lives: amongst other things, they found that one in five girls aged 10–15 are increasingly unhappy with the way they look. There is a clear correlation between this and the use of social media: research published by the American Psychological Association found that young people who reduced their social media use by 50% for just a few weeks had significantly better feelings about their weight and appearance than those whose time online stayed the same [2].

Affirming their sexual identity is a significant life task for adolescents. For those who identify as other than heterosexual, this can be a time of great stress and potential conflict. Although there has been increasing acceptance of homosexuality during the last two decades, there is still a global divide [3]. In 22 of 34 countries surveyed, younger adults are significantly more likely than their older counterparts to say that homosexuality should be accepted by society.

Changes in the adolescent brain also impact development. The prefrontal cortex is where planning, problem solving, and hypothetical thinking happen, and is the last

area to be remodelled in the process called synaptic pruning [4]. Adolescents therefore tend to be impulsive, not think through the consequences of their actions, and be more readily distraught when things do not go well. The environment has an impact on which synapses are pruned and which are strengthened, making experiences during adolescence particularly critical for the development of the individual and who they are becoming.

## 2. The Impact of the Pandemic on Adolescents and Their Wellbeing

While adolescents were physically less vulnerable than others during the pandemic, disruptions to routines, closed schools, increased family stressors, social isolation, and exposure to family violence have led to significant psychological difficulties [5]. Chapter 4 of the UK Government's COVID-19 Mental Health and Wellbeing Surveillance Report [6] outlines findings from a number of sources that present a mixed picture. The TELL study asked one hundred teenagers aged 16–19 to write about their experiences of lockdown [7]. Although many found it difficult, confronting high levels of change, loss, and uncertainty, there were also positive aspects. Despite heightened emotionality, many young people were trying to stay upbeat and care for themselves. Although they had to sacrifice important teenage experiences, they also saw lockdown as an opportunity for growth and development. Feeling connected to others, including the wider society, was a major positive aspect for many, and freedom from pressure was another.

In another study, the Big Ask surveyed nearly half a million children between the ages of six and seventeen about the impact of the pandemic [8]. Although most felt they were doing well, there were particular concerns for the mental health of girls aged 16–17, including an increase in reported eating difficulties. Young people in late adolescence (aged 17–19) who were identified as having emotional difficulties pre-pandemic reported experiencing higher levels of stress, conflict, loneliness, and lower levels of perceived social support than young people without emotional difficulties. However, among younger adolescents aged 10 to 16, there was a more nuanced picture; some who had struggled pre-pandemic showed improvements in their mental health, and others who were initially doing well had increased mental health problems. Going back to school was a trigger for some who started to self-harm again or had suicidal thoughts. Young people with Special Educational Needs and Disabilities (SEND) were especially adversely affected by the pandemic as well as those from disadvantaged backgrounds. Poverty, cramped housing, lack of spaces to exercise, and reduced access to technology contributed to inequality in homeschooling as well as overall wellbeing.

The effects of the pandemic are complex and will be emerging for some time. As many life trajectories are established during adolescence and young adulthood, the Australian Institute for Health and Welfare [9] recommends ongoing comprehensive monitoring, including:

- Wellbeing, including mental wellbeing;
- Access to educational choices after secondary school;
- Education attainment, achievement, and outcomes;
- Longer-term outcomes for young people; for example, the potential consequences of unemployment on their long-term prospects and finances, access to secure housing, and mental health;
- The longer-term impact of COVID-19 on child protection services;
- Experience of domestic violence;
- Variation in outcomes for different population groups.

The impact of COVID-19 on family life also had consequences for young people effectively locked into households. Unsurprisingly, caregivers with more psychological distress themselves report the same for their children [10]. These young people are more likely to argue with the rest of the family and also be more dependent on the caregiver.

The Office for National Statistics [11] reported that 41% of parents felt that the wellbeing of the child or young person in their care was negatively affected by trying to continue education at home, while 33% reported that this was also negatively affecting the rela-

tionships between members of the household. Many families did not feel they had the skills or knowledge to homeschool effectively and were reluctant to force their child to do schoolwork.

The impact of school closures, and the consequent disruption to education, undermined both mental health and future school attendance. Post-pandemic, there has been a marked decrease in students going to school in the UK, with the greatest concern in the secondary sector, where pupils are missing more than 9% of classroom time, compared with an average of about 5.4% in the five years between 2014 and 2019. In total, 1,595,582 pupils in England's primary and secondary schools had missed a significant number of school days in the 2021/22 school year [12]. This is out of total of just under 7 million pupils.

In January 2021, Young Minds UK [13] recommended that wellbeing be made a priority in schools and any measures related to students "catching up" be treated with caution, so as not to add further pressure. Ring-fenced funding should be made available to commission mental health support in schools. They acknowledged that facilities such as youth clubs protect mental health, and that, in time, a network of community early intervention hubs would provide access to mental health support in a non-medicalised setting. There is currently little evidence that this is happening in the UK, and there is concern that mental ill health in children and young people is continuing to escalate, with most of the emphasis being put on individual treatment rather than proactive prevention.

## 3. The ASPIRE Principles

ASPIRE is the acronym for Agency, Safety, Positivity, Inclusion, Respect, and Equity. These principles, identified and supported by decades of positive psychology research, apply not only to wellbeing for individuals but also to systems—community, organisational, educational, and family. They underpin healthy relationships, mental health, and prosocial behaviours. Although these principles stand independently, it is the interaction and application of all six that makes the most difference to how young people might flourish post-COVID-19 in an educational setting.

Agency is self-determination, having a voice, and being able to make decisions that concern you; not being controlled by others. Safety is both physical and emotional, freedom from fear, and being able to trust that others have your wellbeing at heart. Positivity is being strengths and solution focused, as well as promoting the positive emotions and positive actions that enhance the quality of our lives. Inclusion is linked to the importance of belonging and mattering—feeling both valued and being of value. Respect is treating others as you would wish to be treated, with empathy, consideration, and dignity. Equity focuses on being fair and flexible in order to maximise opportunities for everyone.

The following paragraphs show how these principles are being enacted across the world post-COVID-19, especially within school contexts and learning environments. They interact with each other across the ecology of school systems, from the micro-moments of relationships to conversations in the staffroom, the wellbeing of teachers, and the development of policy. Here, we explore how these principles might support adolescent learning and mental health upon returning to school after the pandemic.

## 4. Agency

Young people are not passive in relation to their environment; they are active participants. The more opportunities they have to make decisions and be involved in what is happening around them, the higher their wellbeing is likely to be. Self-determination theory, originally formulated by Edward Deci and Richard Ryan [14], has three components related to psychological needs: autonomy, competence, and relatedness. Autonomy is where a person is the originator of their actions, making choices congruent with their beliefs and values. Their behaviour is not controlled either actively (being forced) or passively (to please others or gain rewards) and is therefore intrinsically motivated. The definition of competence in self-determination theory is having a perception of effectiveness. Feeling positively connected with others fulfils the need for relatedness.



School and work closures meant that young people spent more time at home with family, and as a result, many relationships were "re-set" beyond the natural ones that occur in adolescence. In some circumstances, the challenges of the pandemic were ameliorated by greater closeness and support, whereas other families experienced higher levels of conflict [15].

The young people who reported positive experiences during the pandemic were those who were encouraged by their parents to not focus on what was out of their control, but to look to what they could do, such as following a daily routine, pursuing interests, and spending time outdoors [16].

A Nuffield Foundation project invited 70 young people aged 14–18 in Italy, Lebanon, the UK, and Singapore as co-researchers exploring aspects of the pandemic on their lives [17]. In the end, there were 64 participants. While not exclusively focused on family relationships, the study revealed young people's insights into the impact of the pandemic on family dynamics and their own roles in the family. The data were analysed under three headings: the family as a prism mediating young people's experiences of the pandemic; the transformative effects of the pandemic; and young people's agency.

Young people without pre-existing medical vulnerabilities were less likely to catch COVID-19, and as a result took on a greater responsibility within families. This included contributions to the homeschooling and care of younger siblings, attending to the welfare of more frail family members, and being more involved in decision making. In several cases, there was a greater awareness of the everyday lives and responsibilities of parents and a newfound appreciation for this. Some participants found a stronger voice during lockdown and challenged parental views that differed from their own.

Although the pandemic has arguably had more impact on young people and their futures than other sections of the population, there are also indications that it is young people who are leading post-COVID-19 response and recovery.

Developed with and led by young people and youth-focused organisations, Global Youth Mobilization launched in December 2020 with funding support from the World Health Organisation and the Big Six [18]. There have been 471 projects in 125 countries, supporting 800,000 communities. These projects have been developed around four main themes:

- Supporting COVID-19 prevention measures and tackling misinformation;
- Physical and mental health challenges;
- Disruption in education and improving employment prospects;
- Overcome gender inequality and combat domestic and gender-based violence.

The voices of some of the young people involved in these projects can be heard in this video: https://globalyouthmobilization.org/ "We are unstoppable together" (accessed on 8 June 2023).

## 5. What Does Agency Mean for Post-COVID-19 Responses in Schools?

The pandemic and the lockdown were out of people's control. What helped adolescents to deal with this was the ability to regain some control in their lives, and in some instances, experience this more than before. It makes sense, therefore, for schools to explore ways to grant students agency wherever possible.

Agency includes students making choices about what and how they learn. Current educational policy in many countries is often a "one-size fits all" curriculum and a top-down, teacher-led approach to learning. For many adolescents, school is neither a relevant nor exciting place to be. Encouraging students back to school post-pandemic means exploring what will motivate them to be there other than the offer of exam passes, which may no longer hold the appeal of future security in a changed world.

UNICEF Australia are exploring ways to give young people a voice in decision making because it is important that they are involved in shaping the policies and programs that will affect their lives [19]. This could also happen in schools. UNESCO carried out research exploring the impact of COVID-19 on the student voice [20]. Their recommendations

include: building a lively participative culture, monitoring progress in giving students a voice and providing time and resources to develop this, promoting the development of democratic competencies, prioritising the student voice in teacher training, engaging "hard-to-reach" students in student voice projects, and consulting students on educational policy making. Students are encouraged to mentor and support each other and work together to explain to adults why the student voice matters.

When students are asked what is helpful for their learning or wellbeing, they need to know that this will be followed up, and plans based on their ideas will be put in place where possible. Schools with a strong commitment to the pupil voice have reported positive outcomes, which include a reduction in exclusions, better behaviour, better relationships across the school community, and improved attainment and attendance [21].

Many young people are capable, when given the opportunity, of taking initiative, being problem solvers, making decisions, and shouldering responsibility. There are two further ways in which such opportunities might be offered. "Service Learning" is the interaction of knowledge and skills with experience, where young people work within the community and formally reflect on their learning. The International Baccalaureate is recognised across the world as a way of empowering school-age students to take ownership of their learning. It is broader than many other curricula and is focused on benefits to the whole community. The Times Education Commission Report [22] recommends the development of a British Baccalaureate.

## 6. Safety

The very reason measures were taken during the pandemic was to protect the physical health of the population before an effective vaccine could be developed. However, safety has been compromised in several other ways during and after the pandemic.

Lockdowns significantly worsened child abuse across the world, especially in poorer nations. Park and Walsh reported that although there was an initial drop in child abuse referrals and notifications, hospital admissions related to child abuse increased [23]. The authors conclude that since the UN predicted that child abuse will stabilise at a higher level, steps must be taken to detect and repair child protection services. This includes safe-guarding procedures in schools.

According to Digital 2020, social media use increased by about 10% during lockdowns as this was considered a safe space for social interaction and entertainment, and it is now estimated that over half the globe is using the internet [24]. However, for some individuals, this also presents risks to both safety and healthy development. In their analysis of the impact of COVID-19 on gambling, for instance, Hodgins and Stevens found that, during the lockdown problem, gambling become more severe, reaching younger age groups, and boys were more at risk [25]. Social media also exposes users to multiple images that present idealised beauty, unattainable except by a very few.

UNWomen called the increase in violence against women during COVID-19 "the shadow pandemic" [26]. It has been exacerbated by economic issues both within families and society, where resources in some countries have been diverted away from the provision of safe refuges. Issues with misogyny and toxic masculinity have hit the headlines many times in recent years, and adolescents living with such values will have been witness to coercive control and domestic violence. Close proximity to stereotyped role models will have impacted gender identities. The prevalence of pornography strengthens attitudes of sexual violence and debasing women, as well as promoting unsafe sexual practices [27].

Young people who are not regularly in school are not only reducing their chances of gaining qualifications but are also vulnerable to risk. The United Nations Office on Drugs and Crime estimates that over one billion young people, or 60% of all enrolled learners, were affected by school closures globally [28]. Adolescents are, therefore, now at a higher risk of exploitation, violence, and abuse, including recruitment by terrorist and other criminal networks [29]. These may offer promises of easy money, leading teenagers into criminal and sometimes dangerous activities. With limited access to education, commu-

nity services, and sports, young people may turn to a range of negative coping mechanisms such as alcohol, drug use, and self-harm.

## 7. What Does Safety Mean for Post-COVID-19 Responses in Schools?

If students are in school or otherwise engaged in supervised activity, they are safer. Making education relevant, engaging, and enjoyable, as well taking place in a safe environment, may encourage young people back into learning. Blomfield and Barker found that extracurricular participation was positively associated with higher academic enrolment, university aspirations, and school belonging, and negatively associated with skipping school [30].

In the UK, schools are required to follow the government's guidance on safeguarding [31]. A "safeguarding lead" is responsible for keeping children safe from harm. This role incorporates prevention, early intervention, monitoring, and response to issues of concern. All staff, however, are expected to provide a safe environment where children can learn, and to identify any student, of whatever age, who is in need of "early help". At all times, the "best interests of the child" are paramount.

Although schools have traditionally been places of safety for young people, this has not been true for everyone, and there have been several reports of students being relieved that they did not have to go during lockdown. Two reasons for this are not being faced with constant failure in a predominantly academic milieu and being bullied by peers.

A safe environment is where students are not fearful of failure. Competition is rarely motivating for those who constantly see themselves as losers. Both are associated with negative emotions such as embarrassment and shame, but this can be reduced when collaboration is increased, personal bests are introduced, and mistakes are positioned differently as an essential element of learning [32].

Bullying behaviours are still rife in secondary schools, with verbal, physical, relational, and/or cyber-bullying affecting up to half of students in many countries. In order to promote greater safety, social and emotional learning (SEL) promises the best outcomes [33]. According to the Committee for Children in the US [34], SEL helps prevent bullying by:

- Teaching relationship skills, including peer support;
- Cultivating empathy to both reduce bullying behaviours and increase bystander intervention;
- Fostering positive classroom norms endorsed by all students;
- Strengthening social problem-solving skills, which can prevent both bullying and victimisation;
- Supporting emotion-regulation skills.

SEL, with an appropriately safe and solution-focused pedagogy, provides students with opportunities to discuss and reflect together; it also has the potential to address other issues that undermine safety, such as addictions, social media use, toxic masculinity, and racism [35,36].

## 8. Positivity

It is hard for many to feel positive about themselves, others, or the world around them following the pandemic. The World Happiness Report 2023 found that young people were increasingly unhappy, with an escalation of mental ill health evident across the globe [37]. Although acknowledging that the negative validates people's experience, it is also valuable to explore what supports optimism, hope, coping skills, and positive emotions. Waters and colleagues summarised positive psychology research, which identified what might buffer, bolster, and build mental health in a pandemic [38]. Buffering is what helps reduce or inhibit psychological ill health, a bolstering effect is what might be put in place to maintain good mental health despite crises, and building is using the challenging experiences to develop new, positive practices and processes. The authors found that distress and wellbeing can coexist, sometimes with one building on the other—such as being grateful for what you have rather than focusing on the challenges and losses.

### 9. What Does Positivity Mean for Post-COVID-19 Responses in Schools?

There are two strands to positivity for adolescents in a school setting to promote positive cognition, words, feelings, actions, and attitudes. The first is individual students learning about resilience and effective coping skills as part of the curriculum, and the second is whole-school intervention at a systemic level. Wellbeing strategies such as relaxation, mindfulness, and gratitude are often taught as part of a universal positive education curriculum, and not specifically for students who have experienced adversity. When these are already established ways of being for students, they are helpful when challenges do arise. Dray and colleagues reviewed 49 studies on resilience-focused intervention programs for children and adolescents, and found they were effective in reducing depressive symptoms, internalising problems, externalising problems, and general psychological distress [39]. There is also a growing body of research on post-traumatic growth that explores the benefits of specific interventions for young people who have struggled with trauma [40,41]. These are likely to be applicable in supporting students post-COVID-19.

Maladaptive coping strategies include taking substances to feel differently or disengaging with activities perceived as stressful [42]. Adaptive coping strategies, such as positive reappraisal, emotional processing, and strengths use, have been found to inhibit post-disaster psychological problems: positive reappraisal involves reframing a situation from unmitigated disaster to focusing on anything good that might have come out of it [43]; emotional processing is the technique of actively processing and expressing one's emotions during times of stress instead of shutting them down [44]; and strengths are those positive personal qualities and characteristics that energise, engage, and satisfy. Being aware of what these are enables them to be put into action when needed. Strengths use has been found to provide a buffer between violence and post-traumatic stress disorder for young people exposed to lengthy periods of war and political conflict [45]. Gratitude is aligned with positive reappraisal and is valuable for wellbeing. Rather than ruminate on the negative, it encourages young people to consider what they might be thankful for [46]. This in itself promotes hope and optimism for the future. Gratitude diaries are one way of doing this.

Whole-school positivity can be summarised as the quality of relationships between all stakeholders and the level of social capital that ensues [47]. This would include how kindness, consideration, warmth, generosity, and gentleness are threaded through all interactions, modelled by staff and regularly discussed. It is about noticing and acknowledging what is going well, and the efforts that people make. It is using strengths-based rather than deficit-based language, and ensuring that the learning environment is one where students can both make progress but also enjoy learning without the pressure of performance. Noble et al., in their analysis of approaches to student wellbeing, identified seven pathways that made a difference [48]: these included pro-social values, a supportive and caring school community, a strengths approach, and social and emotional learning.

### 10. Inclusion

As adolescents seek to establish their own identity, they routinely shift their reference points for values and aspirations away from family to friends. This is, therefore, an important period for peer connection and social interaction. School closures, social distancing and loss of face-to-face social relationships during the pandemic, therefore, presented particular challenges to teenagers who had to isolate at home during successive lockdowns, often for many weeks. Loneliness has been one of the concerning outcomes for adolescents and their development. In July/August 2020, 42% of young people aged 13–17 in Australia said that the pandemic had negatively affected their social connectedness, although things improved once restrictions were eased [9].

Belonging, however, has two dimensions: inclusive and exclusive [49]. Inclusive belonging is healthy for everyone, whereas exclusive belonging privileges those "inside" the group who see themselves as superior, and develop rules, regulations, and a uniformity that keep others out. Social exclusion can have a seriously detrimental effect on mental

health. Exclusive belonging is at the root of racism, homophobia, entitlement, and other social problems, and underpins xenophobia.

Adolescents are especially sensitive to the social milieu and the influence of peers [50]. This can either be positive, such as providing emotional support, or negative, as in influencing risk-taking behaviours and "group think".

Some were able to maintain contact in lockdowns through social media, but that in itself was not always a positive experience for those who replaced a few close supportive friendships with being popular, and evaluating themselves against images of "success" and "beauty". The prevalence of cyber bullying also increased feelings of rejection for some. These emotions are powerful, regardless of who is doing the rejection and whether it is mild or extreme [51]. Rejection can lead to loneliness; a lack of meaning; poor mental health, including anxiety and depression; and a reduction in pro-social behaviour.

## 11. What Does Inclusion Mean for Post-COVID-19 Responses in Schools?

Belonging is increasingly recognised as a basic human need and vital to healthy relationships [52,53]. It is feeling that you matter to the people around you. It begins at home with attachment to parents and carers in the first few months of infancy and during childhood but remains linked to wellbeing throughout life.

Widnall and colleagues found that young people's willingness to return to school post-COVID-19 was dependent on their connection to school and relationships with peers and teachers [54]. There was, however, a consensus regarding the importance of returning to school to socialise again with close friends and get back to routines.

School belonging is complex. It includes the extent to which young people feel personally accepted, respected, and supported in the school's social environment, but also that they see themselves making progress in learning, so that attending an educational institution has meaning for them. Prilleltensky talked about the importance of individuals feeling that they matter, and he summarised this as the need to balance a sense of being valued by being able to add value [55]. The return to face-to-face rather than online interactions may also be challenging for some individuals who have lost social confidence and need to build and practice positive social skills [56].

Young people need regular opportunities to interact with each other in a safe, solution-focused environment. Social and emotional learning has been downgraded in some countries, but is increasingly recognised as critical for young people to become more connected, share positive strategies for coping, and learn about each other to reduce prejudice and discrimination. Martinesone and colleagues found that adolescents who experienced greater development in social emotional learning also experienced a greater increase in resilience and prosocial behaviour and a decrease in difficulties [57]. In Circle Solutions, students are mixed up out of their usual social groups to talk with each other about a wide range of issues that are not designed around problems, and they are therefore less likely to raise negative emotions [36,58]. Exploring solutions together builds optimism, enhances communication skills, and promotes connection and shared humanity.

## 12. Respect

Respect can be encapsulated by the Golden Rule: treating others as you might want to be treated yourself, with consideration, empathy, kindness, honesty, and dignity. It is about the quality of relationships from an interpersonal level to how this can impact on policies at the sociopolitical level.

Respect for human rights was challenged during the pandemic as people's freedoms were curtailed. Many, however, readily acceded to the imposed measures as being respectful towards others and their safety. The common good was prioritised before individual freedoms. This was demonstrated by communities supporting each other and valuing keyworkers, but less so by some politicians and those who were profiteering from identified needs.

There are some indications that the pandemic led to increased respect for teachers, as families attempted to homeschool their children and began to fully appreciate the skills of the profession, as well as their efforts to keep schools open for the children of key workers and those with special needs [59].

## 13. What Does Respect Mean for Post-COVID-19 Responses in Schools?

Respectful interactions are not costly, but they are a choice. Being courteous, not jumping to conclusions about someone's motivation, checking out issues that might be interfering with learning, and asking authentic questions are ways of being that enhance positive relationships and make people feel more valued. Behavioural challenges have increased since the return to school, so, more than ever, students need role models around them to demonstrate what a respectful relationship looks and sounds like. This needs to include all interactions across the school with families, colleagues, and support staff. A raised voice is rarely respectful, as is interrupting and speaking over someone, or treating their ideas as worthless. Where teachers are under the pressure of curricular demands, staff shortages, and lack of support, a respectful response to a challenging young person may, however, not be a priority and harder to execute. Respect for teachers and their wellbeing has impact across the system.

Hattie, in his seminal meta-analysis of effective education, states that the quality of teacher–student relationships has a significant impact on learning outcomes [60]. This includes having respect for students' ideas, and listening to them when they talk about how they are making sense of what is being taught.

The corollary of student voice, discussed in the section on Agency, is adult listening. This is both for individual students and across a whole school. When young people are provided with opportunities to share their views, these need to be valued. Teachers require time as well as encouragement to listen to students, so this needs to be built into both pedagogy and priorities. Treating young people with dignity and respect means honouring their experiences and not only being prepared to listen to what they have to say but giving them opportunities to speak. Active listening includes validating emotions and acknowledging that the situations they are facing are significant for them.

Respect is aligned with contextual understanding for both schools and communities [61]. It is acknowledging and taking account of what is happening for people in their particular setting, whether that be loss, poverty, or mental health concerns. A one-size-fits-all approach to young people post-pandemic is not helpful. Young people are not the same, and they have different needs, interests, strengths, and ways of learning. Respecting diversity matters [62] but is hard to deliver when identical expectations are put on all students and all educators.

Respect for the communities that the school serves means finding out what is important to them, and ways in which their culture can be visible. When students do not read, see, or hear about people that look or sound like them, they are less likely to feel that the school is for them.

Respectful relationships include the way people are spoken with, but also about. The culture of a school is dependent on how stakeholders are positioned. Conversations are critical to this. This entails both how individual teachers discuss students and families and the priorities set by the school executive.

Respect means belief in someone—not giving up on them. This is now included in the research on resilience as well as numerous anecdotes about how individuals became a success because of the belief of a teacher [63].

Acceptance is also integral to respect. When students are told that even when their behaviour is not acceptable they are still wanted and valued for themselves, this increases the chance of addressing issues effectively. When adolescents have experienced trauma in the pandemic, they may react to any perceived threat with an emotionally charged response, which may be difficult for those around them. Teachers need to know how to

de-escalate a tense situation with gentleness and respect. Such situations can be alarming for young people as well as others.

It has been especially challenging to be a respectful school leader after the pandemic. Coping with staff absence and attrition, as well as a rise in mental health, behaviour, and attendance issues for students, has often been compounded by government directives that aim to "go back to normal" as soon as possible. Emotionally literate leadership includes acknowledging effort as well as achievements, making each person in the school feel they matter to the organisation, consulting widely and authentically on policies and policy changes, communicating clearly about final decisions, staying calm in a crisis, and apologising when getting things wrong [64].

## 14. Equity

While the pandemic has demonstrated that we are an interconnected humanity, the disruptions brought on by the pandemic are exacerbating inequalities both within and across countries, especially for girls, those with special needs, and already-deprived communities.

Young Lives is a longitudinal study of poverty and inequality that has been following the lives of 12,000 children and young people in Ethiopia, India, Peru, and Vietnam since 2001 [65]. In Peru, the focus is on the ongoing economic and social consequences for young people who have seen their future hopes diminish, their education disrupted, and their opportunities lost [66]. This has had a negative impact on their emotional wellbeing. Although remote education has emerged as a potential useful tool, there is continuing dissatisfaction with the quality of education overall, alongside traditional gender roles which disadvantage female aspirations. Vietnam was one of the first countries to recognise the severity of the COVID-19 health crisis and quickly closed schools and universities. UNICEF estimated, however, that 93% of teachers in remote provinces of Vietnam did not have access to modern technology prior to the pandemic [67]. Educators were, therefore, left to implement online learning with very little experience or expertise. Household finances took a hit, and minority ethnic groups were hardest hit.

Many young people were reliant on online learning, but not everyone was able to access this. UNESCO expresses concern that the pandemic will wipe out several decades of progress, especially in addressing poverty and gender equality [68]. They recommend nine areas for public action: these include a commitment to strengthen education as a right and a call to value the teaching profession and the voice of young people, transforming learning spaces, ensuring access to various forms of technology, and promoting scientific literacy so that students can recognise what is evidence-based knowledge and what is not. There is a plea from UNESCO's Commission to protect public education with sufficient financing and renew commitments to international cooperation to prevent further global inequalities. In the renewal of education, human interaction and wellbeing must be given priority.

In the UK, it has become clear that COVID-19 has increased levels of poverty significantly in low-income families [69], putting even more pressure on single parents. With everything becoming more expensive, families with children are accessing food banks in larger numbers than ever. Even for those in paid work, income has become more tenuous, and low-income working households have been disproportionately affected. Cramped living conditions with little outdoor space have exacerbated problems, including mental health problems, especially in teenagers and young adults. Parents report that older children understand what the pandemic means and worry about the impact on their future. Other populations of increased inequality are those living in the north of the country, belonging to an ethnic group, living in unsafe homes with an increased risk of violence, or with special educational needs [70].

## 15. What Does Equity Mean for Post-COVID-19 Responses in Schools?

Equity can be summarised as flexibility and fairness. There is lip service to "levelling up", but very little is being put in place to ensure that those who need more support are getting it, especially young people [71].

According to the National Foundation for Educational Research (NFER) [72], what young people need in education post-COVID-19 is greater access to learning enrichment, rather than a focus on academic "catch up" and more cognitive overload. This should include physical activity, creative activities, and opportunities for outside visits. Youth services have been decimated in many areas but provide experiences that have the potential to support a young person's personal, social, and educational development [73]. School leaders serving deprived areas need the funding, support, and autonomy to make decisions in the best interests of their pupils. NFER recommend reviewing support for individuals already identified with Special Educational Needs and Disability and identifying others who may be at risk.

One major lesson learned from the pandemic was the value of key workers, and the realisation that it is not only academic brilliance that we need to build a brighter future; rather, we need to acknowledge and credit the skills that are required for everyday living and working. The OECD call for rethinking education that accredits a wide range of knowledge and skills, focuses on life-long learning, and teaches critical thinking, enabling young people to tell the difference between fact and opinion [74].

There is no doubt that the education of young women has been adversely affected by the pandemic globally. Older girls especially are less likely to return to school. The World Economic Forum recommends a range of incentives for girls to attend school, including extra support, the provision of basic needs, from food to hygiene products and cash to families [75].

## 16. Conclusions

This paper has explored the impact of the pandemic on adolescents; their development, their education, and their future. Using the acronym ASPIRE, we have identified why each of these principles matter for wellbeing and ways in which they might be enhanced in educational settings. We acknowledge the complexity of addressing these issues across an ecological framework in schools but offer the following as a brief summary of considerations going forward for policy and practice.

Agency: COVID-19 not only undermined physical health; it also led to a deterioration in mental health. Self-determination is linked to wellbeing, and young people did not always feel they had any control over what was happening. Many international organisations, such as UNESCO, have argued for giving students more of a voice and increased agency to make decisions about matters that affect them. They need opportunities to demonstrate innovation and creativity.

Safety: In order to function well, people need to be and feel safe. Teenagers who are not in school are vulnerable to higher risk of harm, including sexual and criminal exploitation. Encouraging students back to school means ensuring they are welcomed and protected from bullying by peers or humiliation in the face of academic failure. Teachers may need to be relieved of the pressure to achieve high academic grades for all.

Positivity: Despite the devastation caused by the pandemic, young people need to have some optimism and hope for the future. Schools can do much to support mental health by helping students develop protective attitudes and skills that foster resilience. High levels of social capital across a school also build the warm and trusting relationships that support both wellbeing and learning.

Inclusion: Young people often rely on each other to discuss and practise different ways of being—developing a firm identity to help them navigate their adult lives. The pandemic stopped people from meeting up in real time, and although social media was valuable for some, many teenagers reported feeling lonely and out of touch with friends. Having a sense of belonging to a school can be summarised by being valued, being able to contribute value, and seeing progress in learning.

Respect: This is reflected in the consideration of context and circumstance for both individuals and communities. Schools that practice respect do not jump to conclusions about a young person or their behaviour but seek to understand what is contributing to

difficulties and ways to resolve these together. The same applies to families. It means asking good questions and listening to the answers.

Equity: The pandemic accelerated inequalities everywhere: gender, social and economic circumstances, special needs, disability, and location. Social mobility is at risk of stagnation. If education is going to be the driver for increasing opportunities for all as it has been in the past, there needs to be greater fairness in resource allocation and more flexibility within schools to meet diverse needs. This will require political will as well as courage from school leaders.

A report for UNESCO [68] notes that the pandemic has not only revealed vulnerabilities across the world, but also human resourcefulness and potential. They conclude that action must be taken on the 2030 Agenda for Sustainable Development, based on both scientific evidence and a humanistic vision of education, development, and human rights frameworks. They ask that world leaders commit to strengthen education as a common good. In education, as in health, we are safe when everybody is safe; we flourish when everybody flourishes.

The Times Education Commission reports that a third of students in the UK leave school without qualifications and that school needs to be a place that is relevant for them [22]. Education should not be something that anyone can fail in. This document recommends far-reaching changes to pedagogy, curriculum, and the learning environment—beginning with a focus on the early years—that makes the most difference to future attainment. This revisioning of education, however, is some way off, even though places such as Finland and Estonia are making great strides in developing more child-centred education that builds a society fit for the 21st century—enhanced by the knowledge we gained from the pandemic.

**Funding:** This research received no external funding.

**Institutional Review Board Statement:** Not applicable.

**Data Availability Statement:** Not applicable.

**Conflicts of Interest:** The author declares no conflict of interest.

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
