# Peer review of "ASPIRE to a Better Future: The Impact of the Pandemic on Young People, and Options for Schools Post-COVID-19"

_education, doi:10.3390/educsci13060623_

Round 1

Reviewer 1 Report

The paper titled " ASPIRE to a better future: the impact of the pandemic on young people, and options for schools post-Covid in supporting adolescent development, mental health and healthy positive relationships" provides a comprehensive overview of the challenges faced by adolescents globally during the COVID-19 pandemic. The authors employ the ASPIRE framework, focusing on Agency, Safety, Positivity, Inclusion, Respect, and Equity, to analyze the repercussions on various aspects of adolescent development. While the paper is well-written, it is noted to be highly theoretical. From my perspective, the title is quite long and should be shortened.

While the paper presents a theoretical framework, it would benefit from incorporating empirical evidence to support the claims made. Including references to relevant studies, surveys, or interviews conducted with young people regarding their experiences during the pandemic would enhance the paper's credibility. This would also allow readers to connect with the material on a more practical level.

This themed issue of “Education Sciences” seeks articles that explore how we might better support improved mental health and wellbeing in schools, and how we might better encourage relevant stakeholders to adopt effective systemic development as we enter a post-pandemic era in history. The authors didn't put the usual subdivisions in a paper (such as Methods, Results, ..), but to do their analysis the authors must have done at least a literature review. How it was performed?

While the theoretical framework forms the foundation of the paper, it is important to strike a balance between theoretical discussions and practical applications. Consider restructuring the paper to allocate sufficient space for discussing real-world implications and solutions. This will make the paper more accessible and applicable to a broader audience.

Lastly, conduct a thorough proofreading and editing pass to address any grammatical errors, typos, or inconsistencies in formatting. This will ensure that the paper meets the highest standards. In my opinion, the first sentence of the abstract is a bit confusing.

Overall, this paper provides a comprehensive theoretical framework for understanding the impact of the COVID-19 pandemic on young people's development. Its acceptance will depend on the type of papers that the editors want to have in this issue.

Author Response

Thank you for your review.

1. I have shortened the title as suggested.

2. The paper contains several references to research with young people eg the Big Ask surveyed nearly half a million children and young people,  the Nuffield Project carried out research with 64 co-researchers aged 14-18 in four countries.  I have, however added a link to the voices of some young people involved in projects. 

3. The literature review was secured by searching a) both studies and statistics for the impact of Covid on adolescents and b) with the addition of the ASPIRE principles and education.  Apart from some seminal studies, searches focused on 2020 and beyond.  Some searches also included 'positive psychology'  As the paper did not follow an empirical research fomat I did not consider this explanation necessary

4.  I believe that the paper does address practical solutions throughout but have now added a summary of these in the conclusion.

5. The paper has been proof-read once again with one grammatical error identified.

6. I asked several people about the first sentence of the Abstract and they found it acceptable so I have left this in.

Reviewer 2 Report

This is a theoretical paper dealing with post-covid requirements posed towards schools in order to empower and strengthen students regarding their educational attainment and well-being. The rationale is based on the ASPIRE principles, acronym for Agency, Safety, Positivity, Inclusion, Respect and Equity.

My major points:

-      In general, the paper is well-written and easy to follow (with the exception of the conclusion, see below)

-          The paper claims to have an international perspective. However, it seems to me that the paper is written heavily from a UK perspective. I recommend either widening the perspective to be truly international or to rewording the perspective to be more precise about that.

-          The conclusion is rather weak. Instead of quoting any reports, I would like to see the paper state more clearly and with some priorization the actions that need to be taken (e.g. 1) … 2) … 3) …) and who is supposed to take these actions and how (i.e. implications of the paper)

Minor issues:

-          Numbering needs revision to be no longer consecutive throughout the paper.

-          Reference 29 needs revision: It should read Bloomfield & Barber

Author Response

Thank you for your review.

  1. I have changed the Abstract to read 'in the UK and elsewhere'.   I hope this is now acceptable.
  2. I agree that the conclusion was weak.  I have now written a more extensive summary of actions to be taken.
  3. The numbering followed the guidelines given so this remains unchanged.